# Maternal perception, barriers, and facilitators regarding oral rehydration salt solution in diarrhoeal disease: A qualitative study in Bangladesh

**Md Ridwan Islam**[1], **Md Fuad Al Fidah**[1*], **Sneha Paul**[1], **Mahfuz-Un Nesa**[2], **Sarashwati Giri**[2], **Syed Jayedul Bashar**[1], **Chowdhury Ali Kawser**[1], **Tahmeed Ahmed**[1,2,3], **Sharika Nuzhat**[2,3,4]

1 Nutrition Research Division, International Centre for Diarrhoeal Disease Research, Bangladesh (icddr,b), Mohakhali, Dhaka, Bangladesh, 2 BRAC James P Grant School of Public Health, BRAC University, Dhaka, Bangladesh, 3 Office of the Executive Director, International Centre for Diarrhoeal Disease Research, Bangladesh (icddr,b), Mohakhali, Dhaka, Bangladesh, 4 Dhaka Hospital, Clinical and Diagnostic Services, International Centre for Diarrhoeal Disease Research, Bangladesh (icddr,b), Mohakhali, Dhaka, Bangladesh

* fuad.fidah@icddrb.org

## Abstract

### Background

Although oral rehydration salt (ORS) solution is a lifesaving management for diarrhoea, its exertion is still surprisingly low among caregivers. As mothers are considered to be the primary caregivers, they bear the responsibility of administering medications to their children. We aimed to evaluate maternal perceptions along with the barriers and facilitators in play that are affecting ORS administration among children in Bangladesh.

### Methods

The study was conducted at Dhaka Hospital of the International Centre for Diarrhoeal Diseases Research, Bangladesh through a qualitative approach. In-depth Interviews (IDIs) were conducted on thirty-one mothers of under-5 children reporting to the hospital from February to April 2024. Qualitative content analysis method was used to describe and analyse the transcribed data.

### Findings

The mothers held several misconceptions regarding the administration of ORS. Majority of the participants thought their child could receive ORS through their breastmilk, which was an interesting finding. Along with this, mothers also lacked proper understanding regarding ORS preparation, storage, and use. Some of the key barriers were the lack of proper knowledge, motivation, and compliance, along with misdirection

**Data availability statement:** The data underlying this study contain sensitive personal information, including patient identifiers such as name, age, hospital identification number, residential area, and contact details. In accordance with the restrictions imposed by the institutional Ethical Review Committee (ERC), these data cannot be shared publicly to protect participant confidentiality. However, de-identified data may be made available upon reasonable request to qualified researchers who meet the criteria for access to confidential data, as determined by the ERC. Requests may be submitted to the Ethics Committee via Ms. Shiblee Sayeed (email: shiblee_s@icddrb.org)

**Funding:** The author(s) received no specific funding for this work.

**Competing interests:** The authors declare that they have no competing interests.

from the elders and pharmacy drug sellers. Additional key facilitators to ORS use were trust towards doctors, positive attitude towards learning about ORS, community education, instructions written on the packet, and support from the healthcare providers.

## Conclusion

A focused approach should be implemented to educate mothers on the correct preparation and administration of ORS for children, while also addressing and dispelling any misconceptions.

## Introduction

Diarrhoea is considered one of the leading causes of death in the world despite being a disease that can be treated effectively using scientifically proven resources [1]. The Global Burden of Disease Report 2021 pointed out that diarrhoea still causes 15.4 per 100000 deaths each year, making it the 14th leading cause of death globally [2]. When considered against children under the age of five years, it is much more threatening as diarrhoea is the 3rd leading cause of mortality among them, contributing to 9% of all under-5 deaths [3]. It is evident that children residing in low-and middle-income countries (LMICs) suffer more due to diarrhoea associated morbidities compared to the rest of the world [4]. Bangladesh is among the twenty LMICs that share the burden of high mortality rates due to diarrhoea-related complications [5]. The disease-specific mortality rate of diarrhoea among under-5 children in Bangladesh was 12.69 per 100,000 in 2019 [6].

Severe dehydration and loss of fluid have always been the main reason behind diarrhoea-associated deaths [3]. Dehydration is defined by The World Health Organization (WHO) as a condition which is brought about by excessive loss of body water [7]. Unfortunately, in addition to being more susceptible to diarrhoeal disease, young children are also more at risk of developing dehydration [7]. This risk can be attributed to a relatively higher percentage (80%) of total body water among them and their inability to access fluids or let their caregivers know of their needs. Moreover, infants also exhibit higher fluid requirements as they suffer from increased insensible fluid losses [7,8].

Fluid and electrolyte replacement have always been fundamental to the case management of diarrhoea [9]. Oral rehydration solution (ORS) is an affordable, comprehensive, and easy-to-use intervention that has been playing a vital role in managing acute watery diarrhoea for decades, effectively restoring and maintaining hydration [10]. First used in 1968 to treat fluid loss and electrolyte imbalance in cholera patients, ORS has since become a global standard [11]. The International Centre for Diarrhoeal Disease Research, Bangladesh (icddr,b), was instrumental in promoting its worldwide use, earning ORS recognition as one of the greatest medical advances of the 20th century [12]. The WHO has been advocating the use of ORS globally since 1978 to manage dehydration in patients with diarrhoea, and its utilization has successfully saved almost 54 million lives since 2007 [13].

Despite its proven effectiveness, the use of ORS remains surprisingly low during patient care in several countries [13,14]. Since the 1990s, the promotion of Oral Rehydration Therapy (ORT), which includes ORS and recommended homemade fluids (RHF) [15], has slowed, leading to its declining coverage [13]. It has been reported that globally only 43% of patients suffering from diarrhoeal disease are treated with ORT [13,16]. The coverage of ORS in LMICs, especially in South Asia and Sub-Saharan Africa, still remains inadequate [17]. In 2021, only 36% of under-5 children with diarrhoea received ORS in sub-Saharan Africa, which is rather alarming [18]. In their systematic review, Ezezika *et al.* reported that approximately 300,000 diarrhoea-associated deaths among children of LMICs could have been avoided if ORS had been provided to them in order to manage dehydration [13]. A wide range of factors is responsible for the low proportion of ORS use despite being a well-known therapy to combat diarrhoea. Issues like insufficient resources, lack of political commitment, and inadequate infrastructure, along with socio-cultural factors such as the negative perception of ORS regarding perceived benefit by the parents, low awareness of ORS usage, etc, are acting as contributing factors to this decreasing coverage [13,14,19,20]. It is apparent that barriers to a certain healthcare intervention can also hamper its administration and implementation [21]. As mothers are the primary caregivers of children in these parts of the world, they are responsible for administering most of the medications to their children. Understanding their perceptions of ORS and identifying potential barriers to the implementation of ORS usage is crucial for improving diarrhoea management plans and ensuring widespread coverage of ORS. Although there are several studies that explore other aspects of ORS, data is limited regarding our objective of this study. So, we aimed to explore the perception regarding ORS along with the barriers and facilitators that might be associated with their use among mothers of under-5 children who are suffering from diarrhoea.

## Methods

### Study design and settings

We followed a qualitative approach where in-depth interviews (IDIs) were conducted among thirty-one mothers whose children were admitted to the Dhaka Hospital, icddr,b. Research location and participants were selected purposively in our study. Dhaka Hospital is the largest diarrhoeal disease hospital in the world that serves patients of all ages in different units according to the severity of the diarrhoeal disease. Short Stay Unit (SSU), Longer Stay Unit (LSU), and Intensive Care Unit (ICU) provide treatment to the patients as required by their clinical condition [22]. Mothers aged≥18 years with an under-5 child who was admitted to SSU, LSU, or ICU were considered eligible to be enrolled in our study.

### Data collection

The IDIs were facilitated between 1 February, 2024 to 30 April, 2024. The study investigators, who are medical professionals, public health specialists with at least a master's degree and had previous experience in conducting IDIs, conducted the interviews. The interview guidelines and pre-interview questionnaire were constructed by a research team who were adequately experienced in diarrhoeal disease, paediatric illnesses, and research methodology. The pre-interview questionnaires collected information on socio-demographic characteristics, whereas the IDI guidelines targeted obtaining information on maternal perception, potential barriers, and facilitators on the use of ORS among the populace. The IDIs were conducted within two hours of the children's admission at the hospital and lasted around 30–40 minutes. The interviews were recorded using audio recorders after acquiring written informed consent from the participants. The IDIs were conducted in a separate room to ensure the anonymity and privacy of the participants. Only the interviewer, the interviewee, and the note-taker were present during data collection. Data were considered to be saturated when the investigators determined that nothing significant was being added through the IDIs. In our study, we employed stopping criteria to determine data saturation. This approach involved concluding interviews until no new codes emerged or when the number of newly generated codes fell below 5%. Additionally, the recurrence of individual codes three to five times was considered a key indicator of saturation, aligning with established qualitative research practices [21,23–26].

## Data analysis

The audio recordings were transcribed in Bengali manually. Qualitative content analysis method was used to describe and analyse the transcribed data [21,26]. Both deductive and inductive approaches were utilized in order to develop a code-book. Research objectives were considered for deductive approach and inductive approach focused on emerging themes. Subsequently, a research team with expertise in treating children with diarrhoea, paediatric disease and public health conducted thematic analysis using interpretivist paradigm. While a fixed algorithm does not govern thematic analysis, we conducted the coding process manually to identify themes that portrayed the perception of ORS along with the facilitators and barriers to its use. Each of the team members engaged in iterative readings of the transcripts, employing an inductive approach to code development. The team collaboratively refined emerging themes through regular analytic discussions to ensure coherence and interpretive depth. This process was designed to promote consistency and enhance the credibility of the analytical outcomes and to reduce analyst bias. Following, these themes and quotes relevant to them were translated into English by two of the members separately and finalized after they were reviewed by all the team members. We did not conduct back-translation process in our study. All data were stored under lock and key by the primary author and numerical identification codes were used to ensure the participant's anonymity. We followed the Standards for Reporting Qualitative Research (SRQR) reporting guidelines in our manuscript (S1 Table).

## Reliability and validity

We used various methods to ensure the reliability and validity of data. Our interviewer's team consisted of members who were familiar with local norms, beliefs, and understandings of participants' expressions. As all of them were public health experts with previous experience, training, and knowledge about the ORS, they were able to establish rapport with the participants and understand their perspectives [27]. To ensure reliability and validity, regular peer debriefings were conducted among the members during data analysis, and relevant feedback was incorporated [21]. Throughout the interview process, the interviewers confirmed what they documented and ensured that the participants were probed for additional information to facilitate confirmability [27]. Internal validity was ascertained by triangulating themes among interviews. The validity of the findings was examined and reinforced by the application of qualitative methods and reporting the statements of the participants in their own words [21].

## Ethical consideration

All procedures involving research, consent of study participants, data analysis, and publication were approved by Institutional Review Board (IRB) of the icddr,b that comprises of Research Review Committee (RRC) and Ethical Review Committee (ERC) (Protocol No: 23138; Dated: 09 January, 2024). Written Informed Consent was obtained from each participant before enrolling them in the study.

## Results

A total of 31 participants were interviewed, where 10 (32.3%) mothers were selected from SSU, 11 (35.5%) from LSU, and 10 (32.3%) from ICU, whose children were admitted at the respective units. The majority of the mothers were aged between 21 and 30 years in our study, where the most prevalent age group of their children was between 7 and 24 months. Most of the mothers were housewives (n = 26) and almost equal number of them were living inside (n = 16) and outside (n = 15) Dhaka. The information derived from the pre-interview questionnaire are depicted in **Fig 1**

### Perception of ORS

**Preparation and storage.** Out of the thirty-one IDIs, eight mothers expressed their insight regarding the preparation and storage of ORS. All of them reported that they do not prefer to store ORS. Seven of them mentioned that as

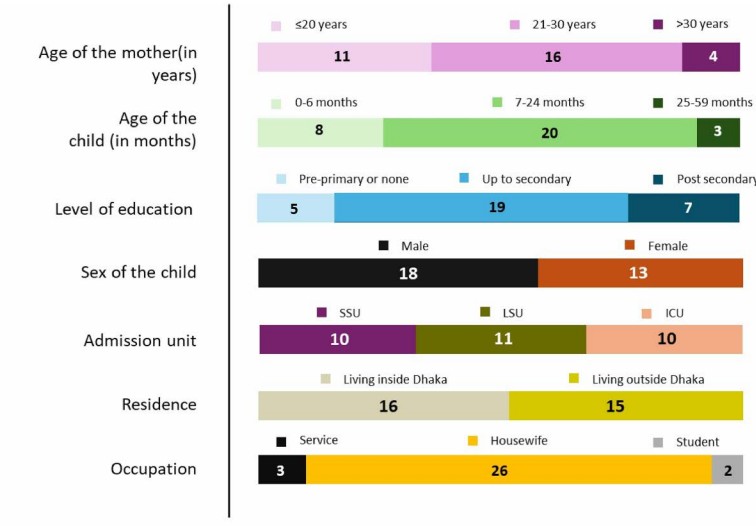

SSU: Short Stay Unit; LSU: Longer Stay Unit; ICU: Intensive Care Unit

**Fig 1. Summary of the socio-demographic characteristics of the participants.**

their baby cannot drink the whole portion of the prepared saline they like to prepare in smaller proportions. This misconception may have led mothers to prepare ORS incorrectly, which can lead to serious health complications such as hypernatremia.

Regarding this issue, a mother commented, *"I take two to three spoons of water, then mix some oral rehydration salt into them according to my assumption. Then I feed the prepared saline to my baby (Participant 30)."*

Concerning the storage of ORS, one mother responded, *"Oral saline turns to poison if we store it for a long period of time (participant 06)".*

**ORS through breastmilk.** Out of the twenty-nine mothers who shared their understanding of ORS and breastfeeding, twenty-three believed that if the mother drinks ORS, it will pass into her breastmilk and can transfer to the child after breastmilk intake. This reflects a critical misconception within the community.

One mother claimed, *"If the baby does not want to drink oral saline, the mother should consume it, the solution will pass through her breast milk and benefit the baby (Participant 28)."* Participant 2 mentioned, *"The elders in my family told me that if I drink the saline solution, my baby will benefit from it by receiving it through my breast milk."*

Three of the mothers said they had no knowledge regarding this issue, and the other three commented that there was no association between ORS and breast milk.

**ORS and antibiotics.** We recorded views from thirty mothers on the use antibiotics over ORS. Although majority of the mothers preferred the administration of ORS to their children during diarrhoea (**Fig 2**), two mothers leaned more towards the consumption of antibiotics.

A participant reported, *"Patient recovers quickly if an antibiotic is provided. We always consulted pharmacy doctors while suffering from diarrhoea and consumed antibiotics according to their advice (Participant 5)."* Here the patient meant drug-seller at local pharmacies by 'pharmacy doctors' who are not registered physicians. Another mother commented, *"I think my baby would have recovered sooner if antibiotics were prescribed (Participant 14)."*

**Side effects of ORS.** Two mothers expressed concerns about the side effects of ORS, stating that consuming too much ORS could be harmful. One mother claimed, *"Drinking too much oral saline is harmful, it can cause problems in the body (Participant 01)."*

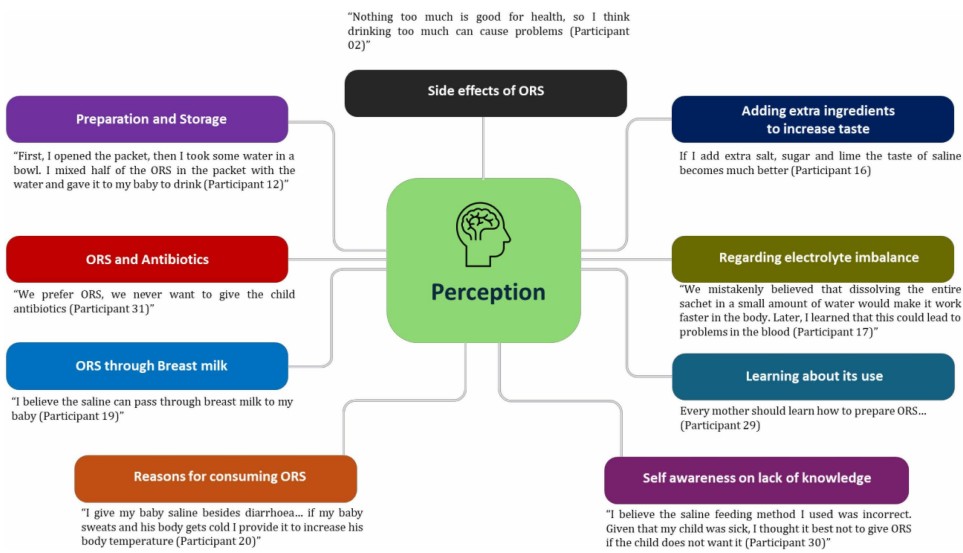

**Fig 2. Relevant quotes illustrating maternal perception regarding ORS use in children.**

**Reasons for consuming ORS.** In our study, mothers stated various reasons for using ORS. They discussed that ORS could be used not only in diarrhoea but also in cases where the baby is suffering from abdominal distension, generalized weakness, hypothermia, vomiting, iron and calcium deficiency, dehydration, and during hot seasons. Additionally, they explained that ORS can also be provided to adults after profuse sweating and a sudden decrease in blood pressure. Mothers further commented that ORS could be applied to quench their thirst, reduce weakness, alleviate any discomfort, and improve vertigo and burning sensation during micturition. One participant said, *"I used to drink ORS during pregnancy as my body became very weak… (Participant 13)."*

**Adding extra ingredients to increase taste.** Five mothers expressed their perspective on the taste of ORS. Two of them claimed that they added additional sugar or lime in order to increase the taste of the ORS. One of them quoted, *"Yes, I added extra lemon and sugar to increase the taste of ORS during the hot seasons (Participant 28)".* Other participants perceived the taste of ORS to be satisfactory.

**Regarding electrolyte imbalance.** Approximately half the participants (n = 15) commented on the elevation of salt content in the body due to improper ORS consumption. Six of them reported that they do not possess a clear conception regarding this issue. Out of the nine mothers who reported that they were aware of this complication, one reflected on her own experience, *"About a month and a half ago I took my child to the 'X' Hospital, near my home. On that day, a child was brought in from 'Y' city. The child had been admitted to a private hospital in that area. Apparently, the nurse there didn't know how to properly mix saline. She mixed a whole saline solution with just a little bit of water and immediately gave it to the child to drink. The child was only eight months old. As a result, both of the child's kidneys were damaged. I personally witnessed this, and it was really heart breaking (Participant 20)".*

**Learning about its use.** In our study, participants agreed that mothers of under-5 children should learn how to prepare and use ORS correctly. One mother reported, *"It is necessary for mothers to learn how to prepare oral saline and when to use it. Because saline is more beneficial to the baby compared to other medicines… (Participant 30)."*

**Self-awareness on lack of knowledge.** A number of participants (n = 6) commented that they are aware of their lack of knowledge regarding the use of ORS. One mother commented, *"I believe, the saline feeding method I used was incorrect. Given that my child was sick, I thought it best not to give ORS if the child does not want it… (Participant 30)".*

Additional significant relevant quotes are illustrated in Fig 2.

## Barriers and facilitators to ORS use

We identified relevant barriers and facilitators of ORS use among mothers of under-5 children who were suffering from diarrhoea. We classified the facilitators and the barriers into three principal domains: the individual level that reflects personal experience while using ORS, the societal level that mirrors factors that emerged from the society participants currently living; and lastly, the policy level which portrays the country-wide practices that might modulate the use of ORS among mothers (Fig 3).

## Barriers to ORS use

Barriers affecting the usage of ORS were identified as factors that prevent or limit the proper utilization of ORS in diarrhoea. Table 1 describes the specific barriers under each of the three domains with illustrative quotes from the respondents.

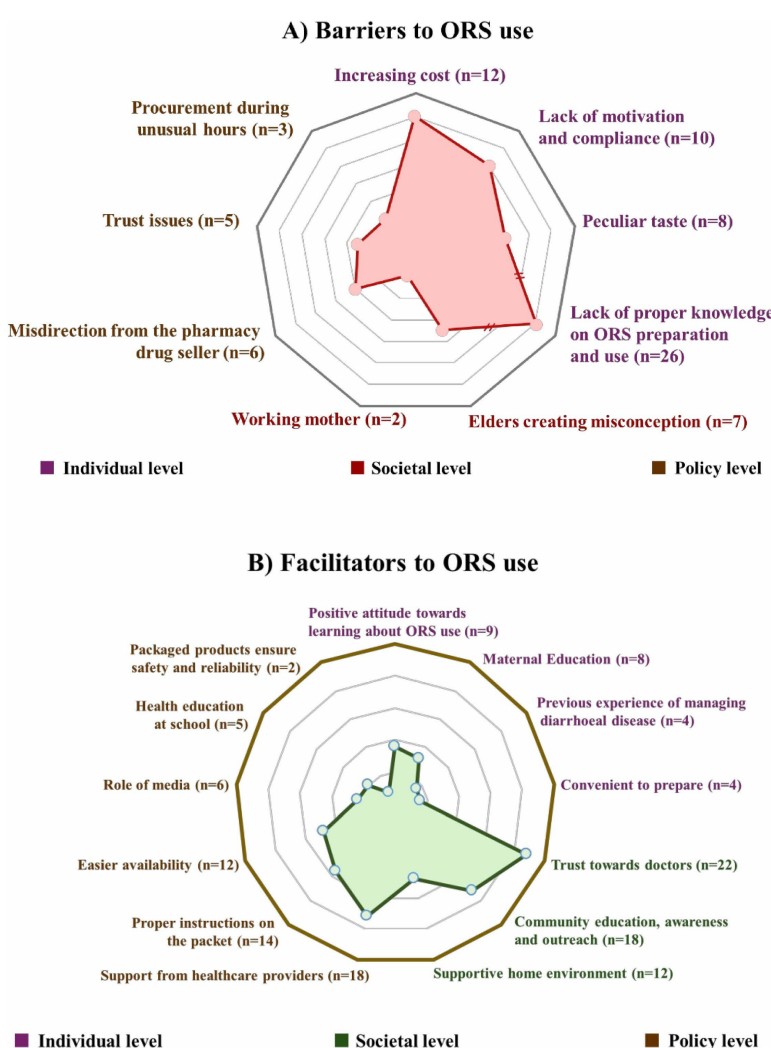

**Fig 3. Emerging themes regarding the barriers and facilitators of ORS use; A) Barriers to ORS use, B) Facilitators to ORS use.**

**Table 1. Relevant illustrative quotes from the participants regarding barriers to the use of ORS.**

| Domain | Specific barriers | Illustrative quotes |
|---|---|---|
| **Individual level** | Lack of proper knowledge regarding ORS preparation and use | *I take a small feeder, then pour water in it. Then, I mix one spoon of saline (ORS) with it to prepare the solution. Then I feed it to my baby. (Participant 2)*<br>*I took half a litre of water, mixed half of the saline (ORS), then set aside the rest. Later, when he passed loose stool after three hours, I discarded the previous saline and prepared another solution with the remaining half again (Participant 31)* |
| | Increasing cost | *There are some who may not be as well-off as me. So, it might be difficult for them to buy oral saline at this high price (Participant 4)*<br>*The price seems very high to me, recently it has increased to six taka per packet (Participant 14)* |
| | Peculiar taste | *My baby does not like the taste of oral saline, I have to feed him forcefully (Participant 2)* |
| | Lack of motivation and compliance | *It is easier to prepare saline at home rather than going to the market. We have sugar and salt at home. We even have Himalayan pink salt. So home-made saline is preferable to me than packaged saline (ORS) (Participant 7)*<br>*Homemade saline is easier to prepare than store-bought ORS, which can be a hassle to purchase at a pharmacy. Plus, I enjoy the taste of homemade saline more… (Participant 23)* |
| **Societal level** | Working mother | *My baby passed stool all day. But I work as a tailor, so I was not at home (Participant 25)* |
| | Elders creating misconception | *My mother used to say that if you have a stomach ache, mix a little saline (ORS) with some water and drink it, and the pain will go away (Participant 4)*<br>*The elders in my family told me that if I drink the saline solution, my baby will benefit from it by receiving it through my breast milk (Participant 2)* |
| **Policy level** | Trust issues | *The packaged saline (ORS) can be duplicate, with home-made saline there is no chance of duplication (Participant 24)* |
| | Procurement during unusual hours | *During late night or early morning, if I don't have any saline (ORS) at home, it is almost impossible to get one (Participant 24)* |
| | Misdirection from the pharmacy drug seller | *The man at the pharmacy (drug-seller) told me that my baby would not be able to consume half litre saline. So, I should take small amount of water and or-saline (local brand name for packaged ORS), then prepare the solution to feed my baby (Participant 21)* |

**Individual level.** We identified four individual-level barriers after analysing the IDIs. The most common deterrent we identified was the participants' lack of knowledge regarding ORS use (n = 26), as demonstrated in their IDIs. The increasing cost of ORS was regarded as one of the most frequent barriers (n = 12) by the patients. Lack of motivation and compliance (n = 10) to use ORS during diarrhoea was another significant barrier. Some of the participants complained that the peculiar taste of ORS (n = 8) contributed to the children's lack of interest in consuming it (Fig 3).

**Societal level.** Our analysis of the interviews revealed that being a working mother can be a barrier to providing ORS to a child who is suffering from diarrhoea, as noted by two participants when they were probed. Another exploration revealed that the elders at the home or community spread misconceptions that actively worked as a barrier to the application of ORS which was reported by seven participants (Fig 3).

**Policy level.** Our study identified three policy-level barriers for the ORS. 'Misdirection from the pharmacy drug seller' was the frequently mentioned policy-level barrier, whereas 'trust issues' against the packaged ORS, and 'procurement of ORS during unusual hours' were also identified as potential barriers (Fig 3).

## Facilitators of ORS use

Facilitators of ORS use were regarded as factors that promote or encourage the widespread adoption and proper utilization of ORS. Table 2 describes the specific facilitators under each of the three domains with significant quotes from the participants.

**Individual level.** We identified four individual-level facilitators after analysing the IDIs. The most frequently mentioned facilitators were 'positive attitude towards learning about ORS use' and 'maternal education'. Convenience of preparation and previous experience in managing diarrhoeal disease were reported as facilitators by four participants (Fig 3).

**Table 2. Relevant illustrative quotes from the participants regarding facilitators of ORS use.**

| Domain | Specific facilitators | Illustrative quotes |
|---|---|---|
| **Individual level** | Convenient to prepare | *It is easy and hassle-free to use the saline that is packaged (Participant 26)* |
| | Positive attitude towards learning about ORS use | *It is necessary for mothers to learn how to prepare oral saline and when to use it. Because saline is more beneficial to the baby compared to other medicines… (Participant 30)* |
| | Previous experience of managing diarrhoeal disease | *I visited this hospital when my first baby was seven years old… he, too, was suffering from diarrhoea. I benefited a lot from that time, which is why I brought my little child here today (Participant 20)* |
| | Maternal Education | *I have studied till fifth grade. I have learned about oral saline from my school… (Participant 13)*<br>*I have learned about oral saline from my school teachers… the instructions are also written on the packet (Participant 31)*<br>*When I was a child, I was taught at school that 3 handfuls of sugarcane molasses and 1 pinch of salt should be mixed with water to make saline. Then I made it just for fun. I still remember it (Participant 26)* |
| **Societal level** | Supportive home environment | *My sister helps me to prepare saline. Even in this instance, when I was staying at my sister's home, she helped me (Participant 18)* |
| | Community education, awareness, and outreach | *In my childhood, my grandparents used to say if you are suffering from diarrhoea then buy and drink ORS. I learned this from them (Participant 14)* |
| | Trust towards doctors | *Doctors say that oral saline (ORS) is more beneficial during diarrhoea, so it is essential… (Participant 30)*<br>*No…I believe in the words of the doctors, or-saline (ORS) is more useful in diarrhoea… (Participant 5)* |
| **Policy level** | Packaged products ensure safety and reliability | *The packet ensures safety. The sachet makes it easier to prepare. Otherwise, the salt and sugar amount may vary (Participant 20)* |
| | Support from healthcare providers | *The healthcare worker at the hospital taught me how to prepare and provide oral saline to my baby… (Participant 27)*<br>*The doctor taught me… when my baby had diarrhoea the last time, we went to a doctor, and he told us to give ORS (Participant 19)* |
| | Easier availability | *Yes, ORS is sold even in the stores of my village… (Participant 15)* |
| | Role of media | *The famous movie star 'X' (name retracted) also acted in an advertisement for oral saline…then there was an advertisement for or-saline (local trade name for ORS), which aired when I was little. I still remember that advertisement… (Participant 20)* |
| | Proper instructions on the packet | *[The process of] preparation is written in the sachet packet. I have read it and learned from the packet (Participant 27)* |
| | Health education at school | *I have first heard about the ORS from school (Participant 23)* |

**Societal level.** After analysing the IDIs, we identified three societal-level facilitators. 'Trust towards doctors' (n = 22) and 'Community education, awareness, and outreach' (n = 18) were the most frequently mentioned societal-level facilitator, whereas 'Supportive home environment' was reported by twelve participants (Fig 3).

**Policy level.** A total of six policy-level facilitators of ORS use were identified in our analysis. Out of them, three were most frequently mentioned, namely 'Support from healthcare providers (HCP)' (n = 18), 'Proper instructions on the packet' (n = 14), and 'Easier availability' (n = 12). The other three, i.e., role of media (n = 6), health education at school (n = 5), and packaged products ensure safety and reliability (n = 2), were less frequently mentioned facilitators (Fig 3).

## Discussion

From 1980 to 2018, the population of the world increased by approximately 70%. During the same time, death due to diarrhoea in under-5 children has decreased from 4.6 million annually to 500,000 [28]. It was reported in 2007 that ORT has single-handedly prevented nearly 54 million diarrhoeal deaths [29]. Additionally, the *Lancet, in their editorial,* commented in 1978 that ORT is perhaps the most important medical advancement of the century [30]. Unfortunately, the promotion of ORT has slowed since the 1990s, leading to a scenario where currently only 43% of patients suffering from diarrhoeal disease are treated with ORT in the world [13,16]. The latest Bangladesh Demographic and Health Survey (BDHS) 2022 report tells us that in Bangladesh, only 76% of children who are suffering from diarrhoea receive ORT [31]. Given this

scenario, it is essential to explore the caregiver's perception of ORS as well as the barriers and facilitators to its use reported by them. In our study, we identified eight crucial domains regarding the mothers' perception, whereas barriers and facilitators to ORS use were also identified under three principal domains, i.e., individual level, societal level, and policy level.

An important finding is that mothers believed their children could receive ORS through breastmilk if they consumed the saline solution themselves. It is a prevalent cultural belief in many parts of the world that a baby's health is directly related to a mother's eating habits [32,33]. Although it is recommended to frequently provide breast milk to the baby, thinking that the child is receiving oral saline through the breast milk could be detrimental to the child [34]. The baby would not receive proper replenishment of salt and water through this technique which may lead to further unfavourable outcomes. On the other hand, consuming ORS by the mothers when it is not indicated can lead to their health problems [35].

In our study, we have found that when asked about ORS preparation and storage, mothers commented that they do not prefer to store ORS. Mothers experienced that their baby could not drink the whole portion outright, so they decided to prepare the ORS in small proportions, which is a wrong and rather harmful technique that can lead to hypernatremia. A study from Burkina Faso also reported similar findings where they reported leftover liquid as a consequence of ORS use [36]. One probable countermeasure can be the development of ORS in smaller packets for the paediatric population, which can be used to prepare smaller proportions of saline, for example, 150 or 200 ml of ORS.

Mothers overwhelmingly supported ORS over antibiotics for treating their children, although previous research documented excessive use of antibiotics for treating diarrhoea [9,36]. Generally, ORS is preferred by caregivers as it is inexpensive, more available, and often considered to provide better value for money [36]. Despite being a safe, easy-to-use management procedure, ORS does have some side effects when improperly administered [37]. Two of the caregivers of our study recognized that ORS does have side effects when improperly used. Moreover, approximately half of the participants also commented that improper ORS consumption may cause electrolyte imbalance. Participants of our study reported a wide range of reasons for consuming ORS. It indicates a lack of knowledge regarding the main purpose of ORS, which is to correct dehydration. Several other studies have also reported such results where mothers or caregivers could not properly mention the reason for administering ORS [36,38,39]. It is necessary for the HCPs to promote ORS so that caregivers are also made aware of the beneficial effects of ORS on children who are suffering from diarrhoea [13,36].

The packaged ORS used for treating diarrhoea does not need additional ingredients. Rather, injudiciously adding ingredients may alter the efficacy of the solution [40]. Even a much earlier study conducted in 1997 reported that adding extra ingredients like fruit juice or powders at home alters the packaged ORS's electrolyte content [40]. Only two of our participants reported adding sugar or lime juice to alter the taste. Our study reported another interesting finding; some of the participants are aware that they lack adequate knowledge regarding ORS preparation and use. Such self-awareness can lead to self-improvement and assist in maintaining a good level of practice [41].

We identified four individual-level barriers to ORS use. Lack of knowledge was the most frequently found deterrent which is supported by the findings of other studies [13,20,36]. A study conducted in Bangladesh reported that 22% of participants did not know when to give ORS, and 43% lacked the knowledge regarding the amount to provide [42]. The lack of knowledge is undoubtedly affecting the practice of ORS among mothers. Some participants in our study mentioned that the cost of an ORS sachet was high, although its price remained the same over the past decade [43]. As ORS alone cannot stop loose watery stools but rather provide management to dehydration, caregivers often consider it to be ineffective; as a result, other treatment options seem more worthy of spending money on [36]. As a result, the total expenditure may have been perceived as much higher by the participants. HCPs play a major role in promoting ORS use; using them to convey the message of ORS's efficacy in treating dehydration can reduce this barrier significantly. Maternal lack of motivation and compliance was also a frequently mentioned individual-level barrier. Peculiar taste and lack of proper

knowledge of ORS preparation and use have also been reported as barriers by the participants in our study, similar to another published paper [36]. They reported that their child does not like the taste of ORS. As adding ingredients at home alters the composition of the packaged ORS, research should be conducted to find out ways to make the taste better, possibly by adding flavours at the manufacturing level to address the issue with the taste.

Various misconceptions which are generally present among the elders of the society were considered as societal barriers towards the use of ORS in our study. Cultural beliefs often affect the use of ORS in Bangladesh which was also reported in a systematic review [13]. A study from India also reported that some common misconceptions regarding ORS hamper its administration in this part of the world, which include that breastfeeding and fluid administration should be stopped during diarrhoea [44]. The healthcare-related decision is often made by the senior members of the family, which can act as a barrier to ORS use, as reported by Ansari *et al.* [45]. Another societal level barrier that was identified in our study is being a working mother. Diarrhoea is a disease where the affected child requires constant supervision, and ORS has to be given after each purging to avoid dehydration. Gopaldas *et al.* reported that mothers who are involved in outside work face difficult situations in properly utilizing ORS during diarrhoea [46]. As an LMIC, it is quite common in Bangladesh that both parents participate in jobs to provide for the family. In this scenario, other family members should come forward to actively take care of the child during his ailment.

The policy-level barriers to ORS use are mostly centred around misdirection from the pharmacy drug sellers and trust issues. Community pharmacists (CPs) constitute an important part of the healthcare delivery system, where they provide treatment for various diseases like diarrhoea [47]. In Bangladesh, mothers often prefer informal care over formal care as it is cost-effective and convenient. Also, it may be so that they discern diarrhoea as a disease that may not need advice from a registered physician. All things considered; community pharmacists are in a unique position to promote the use of ORS in diarrhoea. Unfortunately, studies from Khartoum [47], Qatar [48] and Iraq [49] reported that the proportion of prescribing ORS for acute gastroenteritis by CPs is quite low (0.9%, 1%, and 4%, respectively). Although these were simulated patient studies where the patient was an adult, the picture is largely the same for child patients also.

In their systematic review, Ezezika *et al.* identified distribution problems to be a major barrier to ORS use in Bangladesh [13]. Due to a lack of trust in the distribution chain, issues like mistrust can arise. Additionally, we found that procurement during unusual hours is also a barrier to ORS use. Among the non-users of ORS in the Burkina Faso study, 10% also reported that they did not use ORS, as obtaining it requires investing too much time or traveling a long distance [36]. In our opinion, strengthening the supply chain and promoting homemade ORS during emergencies can address these two barriers.

Although no nationwide study was found during our literature review which explored caregivers' knowledge regarding ORS in Bangladesh, several cross-sectional studies paint a grim picture [42,50]. The lack of knowledge about ORS use was also acknowledged by our participants. Conducting educational campaigns on ORS preparation and administration can further promote the use of ORS in children during diarrhoea. A study conducted on mothers aged 18–35 years found a beneficial impact of such campaigns [51]. A positive association between maternal education and ORS use was reported by several studies similar to our findings [52,53]. Education mothers are more likely to be more aware and seek necessary care during their child's sickness.

The societal level facilitators include trust toward doctors, community education, awareness and outreach, and a supportive home environment. Bangladesh was a pioneer in preventing childhood deaths from diarrhoea, beginning in 1970 with a Communicable Disease Control program to monitor outbreaks [54]. In the late 1990s, the Integrated Management of Childhood Illnesses (IMCI) strategy was introduced, incorporating diarrhoea management with other child health concerns within the Health and Population Sector Programme [54]. Efforts by the government and NGOs, including the establishment of Diarrhoea Training Units and enhanced healthcare workers' skills, significantly increased the people's trust in ORS [54]. Additionally, key therapeutic advances led to the development of Oral Rehydration Salts (ORS), which were widely distributed under the National Oral Rehydration Project from 1981 [54]. Limited initial success in rural areas

prompted BRAC's campaign to teach caregivers to prepare a homemade ORS solution 'Lobon-Gur', until pre-packaged ORS became more accessible under the brand ORSaline [54]. These community education, awareness, and outreach efforts have proven to be useful. As stated earlier, a mother's care-seeking behaviour is often regulated by the elders of the family [45]. So, a supportive family acts as a great facilitator of ORS use.

Packaging and easier availability were two frequently mentioned facilitators in our study. Ezezika *et al.*, in their systematic review, as well as the study in Digre *et al.*, also supports this finding [13,36]. The systematic review found that when the ORS package contains instructional messages, several benefits, like better dissemination of information and higher uptake, can be seen [13]. *Digre et al.* also recommended using juice-box-style packaging or pre-mixing to promote its use [36]. Our analysis also identified that one of the interventions that can significantly affect ORS use is the ease of administration [36]. Media and women's education plays an important role in promoting caregivers care-seeking behaviour in Bangladesh, especially for childhood diarrhoea [15,55,56]. Our results have also identified the role of media and health education at school to be significant facilitators.

We examined the perspectives of the mothers regarding ORS use while focusing on individual, societal, and policy-level domains. In countries like Bangladesh, societal and cultural norms greatly impact the care of a child [13]. So incorporating the societal and policy level domains to examine perception and to identify barriers and facilitators is a strength of the study. Additionally, the study was conducted in the largest diarrhoeal diseases hospital in the world that caters to more than 200,000 patients with diarrhoea, > 60% of whom are paediatric patients. We also ensured theoretical saturation as it increased the possibility of identifying all possible barriers and facilitators [57]. So, a wider picture regarding the current perception of, along with barriers and facilitators to ORS use was drawn. However, the results of this study should be considered in view of its limitations. As the representativeness of the participants was not ensured, the findings cannot be generalized to the whole of Bangladesh. Our study also did not explore and compare the impact of policies suggested by other studies in Sub-Saharan Africa and South Asia.

## Conclusion

Our study found that mothers held incorrect perceptions regarding ORS preparation and administration. Along with these findings, different barriers and facilitators were also identified that demand attention from the researchers, governments, and non-government organizations that are associated with policy-making decisions. To address individual-level barriers, targeted health education campaigns should be implemented to enhance caregivers' perception of ORS, especially regarding its role in rehydration rather than curing diarrhoea. These campaigns, delivered through community health workers, maternal clinics, and schools, should be aimed at dispelling prevalent misconceptions—such as the belief that ORS can be transmitted through breastmilk or that its taste necessitates home modification—while emphasizing safe preparation practices and the importance of timely administration during episodes of the disease.

## Supporting information

**S1 File. Standards for Reporting Qualitative Research (SRQR) checklist.**
(DOCX)

**S2 File. Qualitative interview guidelines.**
(PDF)

## Acknowledgments

The authors at icddr,b are grateful to the Governments of Bangladesh, and Canada for providing unrestricted support to the organization.

## Author contributions

**Conceptualization:** Md Ridwan Islam, Chowdhury Ali Kawser, Tahmeed Ahmed, Sharika Nuzhat.

**Data curation:** Md Ridwan Islam, Md Fuad Al Fidah, Sneha Paul, Mahfuz-Un Nesa, Sarashwati Giri, Syed Jayedul Bashar.

**Formal analysis:** Md Ridwan Islam, Md Fuad Al Fidah, Sneha Paul, Mahfuz-Un Nesa, Sarashwati Giri, Syed Jayedul Bashar.

**Supervision:** Sharika Nuzhat.

**Writing – original draft:** Md Ridwan Islam, Md Fuad Al Fidah, Sneha Paul, Mahfuz-Un Nesa, Sarashwati Giri, Syed Jayedul Bashar.

**Writing – review & editing:** Chowdhury Ali Kawser, Tahmeed Ahmed, Sharika Nuzhat.

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
