## [Decision Letter · Decision Letter 0]

2 Mar 2025

PONE-D-24-55589Maternal Perception, Barriers, and Facilitators regarding Oral Rehydration Salt solution in diarrhoeal disease: a Qualitative Study in BangladeshPLOS ONE

Dear Dr. Al Fidah,

Thank you for submitting your manuscript to PLOS ONE. After careful consideration, we feel that it has merit but does not fully meet PLOS ONE’s publication criteria as it currently stands. Therefore, we invite you to submit a revised version of the manuscript that addresses the points raised during the review process.

Reviewers have raised critical concerns relating to methodology and standards of qualitative study design and reporting including inadequate sample size for which could influence final conclusions and implications. 

This means, addressing the reviewers' concerns will require increasing data points, reanalyzing the data, rewriting the manuscript to suit presentation of qualitative study according the journal's standard (e.g., COREQ checklist), new/modified findings and results etc. 

Thus, addressing all queries will mean a significant modification in manuscript. 

We look forward to receiving your revised manuscript.

Kind regards,

Bismark Dwumfour-Asare, MSc, PhD

Academic Editor

PLOS ONE

Journal Requirements:

2. In the online submission form, you indicated that At icddr,b, we adhere to a strict policy to ensure that data containing identifying patient information is not made available. However, data related to this paper are available on request. Researchers who meet the criteria for accessing confidential data may request it by contacting Ms. Shiblee Sayeed (shiblee_s@icddrb.org) from the Research Administration of icddr,b (http://www.icddrb.org/).

Additional Editor Comments:

Reviewers have raised key and critical concerns relating to methodology and standards of qualitative study design and reporting. All these issues including expanding sample size and restructuring will help the manuscript to meet the journal standards.

This means, addressing the reviewers' concerns will require increasing data points, reanalyzing the data, rewriting the manuscript to suit presentation of qualitative study according the journal's standard (e.g., COREQ checklist), new/modified findings and results etc.

Thus, addressing all queries will mean a total change in manuscript and therefore will require resubmission.

Reviewers' comments:

Reviewer's Responses to Questions

**Comments to the Author**

1. Is the manuscript technically sound, and do the data support the conclusions?

Reviewer #1: Yes

Reviewer #2: Yes

2. Has the statistical analysis been performed appropriately and rigorously? 

Reviewer #1: I Don't Know

Reviewer #2: N/A

3. Have the authors made all data underlying the findings in their manuscript fully available?

Reviewer #1: No

Reviewer #2: Yes

4. Is the manuscript presented in an intelligible fashion and written in standard English?

Reviewer #1: Yes

Reviewer #2: Yes

5. Review Comments to the Author

Reviewer #1: Sample Size Concern: While the manuscript presents an interesting study, the sample size used in the experiments appears to be insufficient. A larger sample size would be necessary to provide adequate statistical power, reduce the potential for Type I and Type II errors, and improve the reliability of the conclusions.

Potential Impact on Statistical Analysis: With an insufficient sample size, the statistical analysis may be compromised, and the findings could be less reliable or generalizable. I would recommend that the authors either increase the sample size or clarify how the current sample size still allows for valid conclusions despite potential limitations.

Future Considerations: The authors should consider revisiting the study design and increasing the sample size in future experiments. This would help strengthen the validity of their results and ensure that the conclusions drawn are robust.

Data Sharing Limitation: The manuscript does not include sufficient access to the underlying data. In order to meet standards for transparency and reproducibility, the data points behind summary statistics (e.g., means, medians, variance) should be made available, either as part of the manuscript or in supporting information.

The manuscript is presented in a clear and coherent manner. The structure is logical, with a well-organized flow of information, and the writing is concise, making it easy for readers to follow the research. The language is appropriate for the intended audience, and the scientific terms are used correctly and consistently.

Reviewer #2: Based on the manuscript titled "Maternal Perception, Barriers, and Facilitators regarding Oral Rehydration Salt solution in diarrhoeal disease: In response to your questions about the manuscript "Maternal Perception, Barriers, and Facilitators regarding Oral Rehydration Salt solution in diarrhoeal disease: a Qualitative Study in Bangladesh" published in PLOS ONE.

ORS solutions save lives by treating diarrhoea but caregivers throughout Bangladesh use them infrequently. The research study identifies barriers such as misconceptions and lack of knowledge along with facilitators including trust in doctors and community education for Oral Rehydration Salt usage. These results carry substantial implications for worldwide public health efforts which focus on improving child health measures through better understanding of diarrhoeal disease management and increasing ORS adoption in LMICs.

The authors have meticulously integrated their findings into existing scholarly work. The authors examine global mortality statistics from diarrhoeal diseases while analyzing historical ORS usage patterns and making comparisons with studies from LMICs. The authors examine prior studies from Bangladesh and additional regions that examine cultural beliefs as well as policy-level barriers and community-level interventions. This research offers a balanced overview of existing literature while providing new qualitative observations.

The research uses qualitative inquiry through in-depth interviews (IDIs) from 31 mothers whose children are under five years old. The research uses thematic analysis to find obstacles and factors that promote ORS utilization. The study outcomes match prior research and are reinforced by direct quotes from participants. The study's conclusions would gain more credibility through future research involving a larger sample size and quantitative validation as well as longitudinal follow-up.

This research comprehensively explains its qualitative methodology and details how IDIs and thematic analysis were applied. The study's reproducibility would benefit from an additional file that describes the interview guide alongside the coding process since no specific algorithm exists. The study does not report any deviations from their established protocol.

The study is well-structured and methodologically sound. The study's impact could be enhanced by providing additional details about intervention methods and suggested policies. The document requires only small adjustments before it can be published.

The manuscript states that icddr,b (International Centre for Diarrhoeal Disease Research, Bangladesh) holds the data which can be requested but remains unavailable to the public because of ethical restrictions. Researchers did not use genomic, proteomic or disease-related datasets in their study.

The research follows established ethical and methodological guidelines. The research received approval from an Institutional Review Board and complies with qualitative research reporting standards. The study lacks explicit references to STROBE guidelines for observational design and COREQ standards for qualitative research.

The study includes detailed information concerning participant selection alongside data collection and analysis methods. The study would benefit from additional documentation about coding methods and theme development to improve reproducibility.

Yes, the paper is well-structured and accessible. The work delivers straightforward explanations of its concepts while presenting its results in a clear and understandable format. A few slight changes to both wording and sentence structure would improve the text's readability.

Increasing the sample size or employing mixed-methods research (both qualitative and quantitative) would improve the study's generalizability. The authors have conducted an equitable assessment of the literature and referenced pertinent research. The review would benefit from a detailed discussion about the comparison of findings with interventions in comparable regions like Sub-Saharan Africa and South Asia.

6. PLOS authors have the option to publish the peer review history of their article (what does this mean? ). If published, this will include your full peer review and any attached files.

**Do you want your identity to be public for this peer review?** For information about this choice, including consent withdrawal, please see our Privacy Policy .

Reviewer #1: No

Reviewer #2: No

---

## [Author Response · Author response to Decision Letter 0]

25 Mar 2025

A point-by-point response to reviewers

Comments from Academic Editor

Journal Requirements:

Response: Thank you for your comment. We have adhered to the journal style requirements.

2. In the online submission form, you indicated that At icddr,b, we adhere to a strict policy to ensure that data containing identifying patient information is not made available. However, data related to this paper are available on request. Researchers who meet the criteria for accessing confidential data may request it by contacting Ms. Shiblee Sayeed (shiblee_s@icddrb.org) from the Research Administration of icddr,b (http://www.icddrb.org/).

Response: Thank you for your suggestion. We have revised our data availability statement accordingly, which is stated below,

“The data underlying this study contain sensitive personal information, including patient identifiers such as name, age, hospital identification number, residential area, and contact details. In accordance with the restrictions imposed by the institutional Ethical Review Committee (ERC), these data cannot be shared publicly to protect participant confidentiality. However, de-identified data may be made available upon reasonable request to qualified researchers who meet the criteria for access to confidential data, as determined by the ERC. Requests may be submitted to the Ethics Committee via Ms. Shiblee Sayeed (email: shiblee_s@icddrb.org).” 

Additional Editor Comments:

Reviewers have raised key and critical concerns relating to methodology and standards of qualitative study design and reporting. All these issues including expanding sample size and restructuring will help the manuscript to meet the journal standards.

This means, addressing the reviewers' concerns will require increasing data points, reanalyzing the data, rewriting the manuscript to suit presentation of qualitative study according the journal's standard (e.g., COREQ checklist), new/modified findings and results etc.

Thus, addressing all queries will mean a total change in manuscript and therefore will require resubmission.

Response: Thank you for your comment. We have addressed the reviewer’s concern regarding sample size, statistical power, and type 1 and type 2 error.

We have also added one of the recommended checklists for the presentation of qualitative studies, i.e., the SRQR checklist, with our submission as a supplementary file, and mentioned on Page 9, Line 159.

Comments from Reviewer # 1

Comment 1: Sample Size Concern: While the manuscript presents an interesting study, the sample size used in the experiments appears to be insufficient. A larger sample size would be necessary to provide adequate statistical power, reduce the potential for Type I and Type II errors, and improve the reliability of the conclusions.

Response: Thank you for your thoughtful and insightful comment. I would like to respectfully draw the esteemed editor’s and reviewers’ attention to the fact that our research was conducted as a ‘qualitative study’ without any quantitative component. As such, concepts such as sample size calculation, statistical power, and Type I or Type II errors, which are inherently aligned with quantitative methodologies, are not applicable to our study.

Extensive literature on qualitative research methods has consistently demonstrated that sample size in qualitative studies is determined by the point of data saturation rather than predetermined sample size calculations 1-4. Unlike quantitative research designs—such as case-control studies, cohort studies, or randomized controlled trials—that necessitate sample size calculations to ensure sufficient statistical power for generalizability, qualitative research does not employ statistical techniques such as t-tests, chi-square tests or regression analyses that require such calculations. Consequently, our study's design and analysis appropriately adhere to established qualitative research principles.

In our study, we reached the point of data saturation after 31 In-depth Interviews. So, this was the adequate sample size for our study. In their systematic review of required sample sizes for saturation in qualitative research, Hennink and Kaiser stated that studies focusing on relatively homogeneous study populations reached saturation within 9-17 in-depth interviews 5. Qualitative research primarily focuses on textual data to explain a phenomenon 5. Within qualitative methodology, saturation indicates a point at which no new properties are revealed about a theoretical construct after gathering more data or does not provide any additional theoretical insight 6. The broader understanding of the concept of saturation is widespread and focuses on sample size assessment. Saturation ensures content validity and indicates the adequacy of a sample for the studied phenomenon in taking into account the depth and nuances of the issues under focus 7. In our study, we employed stopping criteria to determine data saturation. This approach involved concluding interviews when no new codes emerged or when the number of newly generated codes fell below 5%. Additionally, the recurrence of individual codes three to five times was considered a key indicator of saturation, aligning with established qualitative research practices 3,4. A new section has been added to the manuscript elaborating on data saturation in our manuscript for better understanding of the readers (Page: 8, Line: 132-136).

Qualitative research typically emphasizes the depth, richness, and complexity of human experiences rather than statistical inference, which is central to quantitative research designs. In quantitative research, Type I error refers to the incorrect rejection of a true null hypothesis (a false positive), while Type II error refers to the failure to reject a false null hypothesis (a false negative). These errors are tied to the probabilistic nature of hypothesis testing, which is not a primary concern in qualitative methodologies. Instead, qualitative research prioritizes trustworthiness, credibility, transferability, and confirmability as markers of rigor. Qualitative inquiry employs strategies such as triangulation and thick description to ensure the validity of findings rather than relying on statistical measures of error 8. Since qualitative research often seeks to explore meanings, experiences, and social contexts, the notion of rejecting or failing to reject hypotheses based on probability thresholds is generally incompatible with its philosophical foundations.

Comment 2: Potential Impact on Statistical Analysis: With an insufficient sample size, the statistical analysis may be compromised, and the findings could be less reliable or generalizable. I would recommend that the authors either increase the sample size or clarify how the current sample size still allows for valid conclusions despite potential limitations.

Response: Thank you for your valuable comment. In our previous response, we tried to provide a clear explanation regarding qualitative methodology, sample size determination in qualitative research, and the concept of data saturation. We trust that this clarification addresses any concerns the esteemed reviewer may have. Based on this explanation, we conclude that our current study does not require an increase in sample size or additional in-depth interviews.

Comment 3: Future Considerations: The authors should consider revisiting the study design and increasing the sample size in future experiments. This would help strengthen the validity of their results and ensure that the conclusions drawn are robust.

Response: Thank you for your comment. The qualitative study design and sample size have been justified through our previous response.

Comment 4: Data Sharing Limitation: The manuscript does not include sufficient access to the underlying data. In order to meet standards for transparency and reproducibility, the data points behind summary statistics (e.g., means, medians, variance) should be made available, either as part of the manuscript or in supporting information.

Response: Thank you for your suggestion. We have revised our data availability statement accordingly, which is stated below,

“The data underlying this study contain sensitive personal information, including patient identifiers such as name, age, hospital identification number, residential area, and contact details. In accordance with the restrictions imposed by the institutional Ethical Review Committee (ERC), these data cannot be shared publicly to protect participant confidentiality. However, de-identified data may be made available upon reasonable request to qualified researchers who meet the criteria for access to confidential data, as determined by the ERC. Requests may be submitted to the Ethics Committee via Ms. Shiblee Sayeed (email: shiblee_s@icddrb.org).”

Also, our result section does not have any ‘means, medians, variances’ as quantitative statistical analysis was not used in this manuscript.

Comment 5: The manuscript is presented in a clear and coherent manner. The structure is logical, with a well-organized flow of information, and the writing is concise, making it easy for readers to follow the research. The language is appropriate for the intended audience, and the scientific terms are used correctly and consistently.

Response: Thank you so much for your positive feedback.

References:

1 Islam, M. R. et al. Health care providers’ knowledge, attitude, and practice regarding facility-based management of children with severe acute malnutrition in Bangladesh. Food and Nutrition Bulletin 43, 465-478 (2022).

2 Fahim, S. M. et al. A qualitative assessment of facility readiness and barriers to the fac ility-based management of childhood severe acute malnutrition in the p ublic healthcare settings in Bangladesh. Public Health Nutrition 25, 2971-2982, doi:10.1017/S1368980022002014 (2022).

3 Hennink, M. M., Kaiser, B. N. & Marconi, V. C. Code saturation versus meaning saturation: how many interviews are enough? Qualitative health research 27, 591-608 (2017).

4 Guest, G., Bunce, A. & Johnson, L. How many interviews are enough? An experiment with data saturation and variability. Field methods 18, 59-82 (2006).

5 Hennink, M. & Kaiser, B. N. Sample sizes for saturation in qualitative research: A systematic review of empirical tests. Social science & medicine 292, 114523 (2022).

6 Bryant, A. et al. in The Sage handbook of grounded theory 1-28 (SAGE Publications Ltd, 2007).

7 Francis, J. J. et al. What is an adequate sample size? Operationalising data saturation for theory-based interview studies. Psychology and health 25, 1229-1245 (2010).

8 Creswell, J. W. & Poth, C. N. Qualitative inquiry and research design: Choosing among five approaches. (Sage publications, 2016).

Comments from Reviewer 2

Comment 1: Based on the manuscript titled "Maternal Perception, Barriers, and Facilitators regarding Oral Rehydration Salt solution in diarrhoeal disease: In response to your questions about the manuscript "Maternal Perception, Barriers, and Facilitators regarding Oral Rehydration Salt solution in diarrhoeal disease: a Qualitative Study in Bangladesh" published in PLOS ONE. ORS solutions save lives by treating diarrhoea but caregivers throughout Bangladesh use them infrequently. The research study identifies barriers such as misconceptions and lack of knowledge along with facilitators including trust in doctors and community education for Oral Rehydration Salt usage. These results carry substantial implications for worldwide public health efforts which focus on improving child health measures through better understanding of diarrhoeal disease management and increasing ORS adoption in LMICs.

The authors have meticulously integrated their findings into existing scholarly work. The authors examine global mortality statistics from diarrhoeal diseases while analyzing historical ORS usage patterns and making comparisons with studies from LMICs. The authors examine prior studies from Bangladesh and additional regions that examine cultural beliefs as well as policy-level barriers and community-level interventions. This research offers a balanced overview of existing literature while providing new qualitative observations.

The research uses qualitative inquiry through in-depth interviews (IDIs) from 31 mothers whose children are under five years old. The research uses thematic analysis to find obstacles and factors that promote ORS utilization. The study outcomes match prior research and are reinforced by direct quotes from participants. The study's conclusions would gain more credibility through future research involving a larger sample size and quantitative validation as well as longitudinal follow-up.

Response: Thank you so much for your positive feedback. After publishing this qualitative article, the authors will pursue a quantitative study with a larger sample size.

Comment 2: This research comprehensively explains its qualitative methodology and details how IDIs and thematic analysis were applied. The study's reproducibility would benefit from an additional file that describes the interview guide alongside the coding process since no specific algorithm exists. The study does not report any deviations from their established protocol.

Response: Thank you so much for your positive comment. We have uploaded our interview guide as a supplementary file. Details about the coding process are updated as follows: “While a fixed algorithm does not govern thematic analysis, we conducted the coding process manually. Members of the research team engaged in iterative readings of the transcripts, employing an inductive approach to code development. The team collaboratively refined emerging themes through regular analytic discussions to ensure coherence and interpretive depth. This process was designed to promote consistency and enhance the credibility of the analytical outcomes.” (Page: 9, line 148-154)

Comment 3: The study is well-structured and methodologically sound. The study's impact could be enhanced by providing additional details about intervention methods and suggested policies.

Response: Thank you for your positive comment. We have suggested some policies and interventions in the light of our findings. The statement reads: “To address individual-level barriers, targeted health education campaigns should be implemented to enhance caregivers’ perception of ORS, especially regarding its role in rehydration rather than curing diarrhoea. These campaigns, delivered through community health workers, maternal clinics, and schools, should be aimed at dispelling prevalent misconceptions—such as the belief that ORS can be transmitted through breastmilk or that its taste necessitates home modification—while emphasizing safe preparation practices and the importance of timely administration during episodes of the disease.” (Page 30, Line 481-487)

Comment 4: The document requires only small adjustments before it can be published.

Response: Thank you for your positive comment. We have gone through the manuscript and made grammatical corrections.

Comment 5: The manuscript states that icddr,b (International Centre for Diarrhoeal Disease Research, Bangladesh) holds the data whic

---

## [Decision Letter · Decision Letter 1]

13 May 2025

Maternal Perception, Barriers, and Facilitators regarding Oral Rehydration Salt solution in diarrhoeal disease: a Qualitative Study in Bangladesh

PONE-D-24-55589R1

Dear Dr. Al Fidah,

We’re pleased to inform you that your manuscript has been judged scientifically suitable for publication and will be formally accepted for publication once it meets all outstanding technical requirements.

Kind regards,

Prof. Bismark Dwumfour-Asare, MSc, PhD

Academic Editor

PLOS ONE

Additional Editor Comments (optional):

Well done to the authors for taking the pain to address all the reviewers' comments/queries.

However, authors should correct the in-text citations which are placed after full stops.

Also, the verbatim quotations from participants should be put in quotation marks " " as much as possible.

Reviewers' comments:

Reviewer's Responses to Questions

**Comments to the Author**

1. If the authors have adequately addressed your comments raised in a previous round of review and you feel that this manuscript is now acceptable for publication, you may indicate that here to bypass the “Comments to the Author” section, enter your conflict of interest statement in the “Confidential to Editor” section, and submit your "Accept" recommendation.

Reviewer #1: All comments have been addressed

Reviewer #2: All comments have been addressed

2. Is the manuscript technically sound, and do the data support the conclusions?

Reviewer #1: Yes

Reviewer #2: Yes

3. Has the statistical analysis been performed appropriately and rigorously? 

Reviewer #1: Yes

Reviewer #2: Yes

4. Have the authors made all data underlying the findings in their manuscript fully available?

Reviewer #1: Yes

Reviewer #2: Yes

5. Is the manuscript presented in an intelligible fashion and written in standard English?

Reviewer #1: Yes

Reviewer #2: Yes

6. Review Comments to the Author

Reviewer #1: (No Response)

Reviewer #2: comments were addressed from the previous review comments about concerns about dual publication, research ethics, or publication ethics .comments were addressed from the previous review comments about concerns about dual publication, research ethics, or publication ethics .

7. PLOS authors have the option to publish the peer review history of their article (what does this mean? ). If published, this will include your full peer review and any attached files.

**Do you want your identity to be public for this peer review?** For information about this choice, including consent withdrawal, please see our Privacy Policy .

Reviewer #1: No

Reviewer #2: **Yes: ** adinarayana andy

---

## [Editor Report · Acceptance letter]

PONE-D-24-55589R1

PLOS ONE

Dear Dr. Al Fidah,

I'm pleased to inform you that your manuscript has been deemed suitable for publication in PLOS ONE. Congratulations! Your manuscript is now being handed over to our production team.

Kind regards,

on behalf of

Prof. Bismark Dwumfour-Asare

Academic Editor

PLOS ONE